# Expression of Jejunal Taste Receptors in Women with Morbid Obesity

**DOI:** 10.3390/nu13072437

**Published:** 2021-07-16

**Authors:** Laia Bertran, Marta Portillo-Carrasquer, Salomé Martínez, Carmen Aguilar, Miguel Lopez-Dupla, David Riesco, Jessica Binetti, Margarita Vives, Fàtima Sabench, Daniel Del Castillo, Cristóbal Richart, Teresa Auguet

**Affiliations:** 1Grup de Recerca GEMMAIR (AGAUR)-Medicina Aplicada (URV), Departament de Medicina i Cirurgia, Institut d’Investigació Sanitària Pere Virgili (IISPV), Universitat Rovira i Virgili (URV), 43007 Tarragona, Spain; laia.bertran@urv.cat (L.B.); marta.portillo.carrasquer@gmail.com (M.P.-C.); caguilar.hj23.ics@gencat.cat (C.A.); jessica.binetti@gmail.com (J.B.); cristobalmanuel.richart@urv.cat (C.R.); 2Servei Anatomia Patològica, Hospital Universitari de Tarragona Joan XXIII, Mallafré Guasch, Universitat Rovira i Virgili (URV), 4, 43007 Tarragona, Spain; mgonzalez.hj23.ics@gencat.cat; 3Servei Medicina Interna, Hospital Universitari de Tarragona Joan XXIII, Mallafré Guasch, Universitat Rovira i Virgili (URV), 4, 43007 Tarragona, Spain; jmlopezdupla.hj23.ics@gencat.cat (M.L.-D.); david_riesco@hotmail.com (D.R.); 4Servei de Cirurgia, Hospital Sant Joan de Reus, Departament de Medicina i Cirurgia, Institut d’Investigació Sanitària Pere Virgili (IISPV), Universitat Rovira i Virgili (URV), Avinguda Doctor Josep Laporte, 2, 43204 Reus, Spain; mvives@gmail.com (M.V.); fatima.sabench@urv.cat (F.S.); danieldel.castillo@urv.cat (D.D.C.)

**Keywords:** metabolic syndrome, nonalcoholic fatty liver disease, taste receptors, gastrointestinal tract

## Abstract

Nutrient sensing plays important roles in promoting satiety and maintaining good homeostatic control. Taste receptors (TAS) are located through the gastrointestinal tract, and recent studies have shown they have a relationship with metabolic disorders. The aim of this study was to analyze the jejunal expression of TAS1R2, TAS1R3, TAS2R14 and TAS2R38 in women with morbid obesity, first classified according to metabolic syndrome presence (MetS; *n* = 24) or absence (non-MetS; *n* = 45) and then classified according to hepatic histology as normal liver (*n* = 28) or nonalcoholic fatty liver disease (*n* = 41). Regarding MetS, we found decreased expression of TAS2R14 in MetS patients. However, when we subclassified patients according to liver histology, we did not find differences between groups. We found negative correlations between glucose levels, triglycerides and MetS with TAS1R3 expression. Moreover, TAS2R14 jejunal expression correlated negatively with the presence of MetS and ghrelin levels and positively with the jejunal Toll-like receptor (TLR)4, peroxisome proliferator-activated receptor (PPAR)-γ, and interleukin (IL)-10 levels. Furthermore, TAS2R38 expression correlated negatively with TLR9 jejunal expression and IL-6 levels and positively with TLR4 levels. Our findings suggest that metabolic dysfunctions such as MetS trigger downregulation of the intestinal TASs. Therefore, taste receptors modulation could be a possible therapeutic target for metabolic disorders.

## 1. Introduction

Metabolic syndrome (MetS) is a pathogenic condition characterized by the presence of three or more comorbidities, such as central obesity, insulin resistance, hypertension, and hyperlipidaemia, influenced by overnutrition and a sedentary lifestyle [1,2,3]. Obesity is the main causal component of this syndrome; however, its mechanistic role in the disease is not clear [4]. Moreover, it has been reported that patients with MetS present chronic low-grade inflammation [5] and oxidative stress [4] with increased risk of suffering type 2 diabetes mellitus (T2DM) and nonalcoholic fatty liver disease (NAFLD) [2,6].

NAFLD is a major public health concern [7] and has emerged as the most common cause of chronic liver disease [8]. NAFLD is defined as the presence of hepatic steatosis in >5% of hepatocytes in the absence of a secondary cause such as alcohol consumption [9]. This chronic disease covers a wide hepatopathological spectrum, beginning with simple steatosis, which may progress to nonalcoholic steatohepatitis (NASH), cirrhosis and hepatocellular carcinoma [9,10]. The progression into advanced stages of the disease is promoted by multiple insults, such as epigenetic factors, lipotoxicity, toll-like receptor (TLR) activation and intestinal dysbiosis [11].

The intestinal microbiota plays an important role in maintaining physiological, metabolic and enzymatic homeostasis [12]. However, alteration of the components and functionality of the gut microbiota causes intestinal dysbiosis, which is associated with metabolic dysfunctions such as MetS and NAFLD [6,13]. Recent studies have demonstrated that intestinal dysbiosis in patients with MetS causes chronic low-grade inflammation due to the translocation of lipopolysaccharide due to an increase in intestinal permeability [14,15]. Moreover, the disturbed microbiota can produce gut microbiota-derived metabolites that reach the liver through portal vein circulation to induce the formation of reactive oxygen species by hepatic stellate cells, triggering the progression of NAFLD [16]. In this sense, numerous studies have shown that systemic inflammation induced by intestinal dysbiosis is highly associated with the development of NAFLD, confirming the need to characterize new gut-liver axis interactions [17].

Nutrient sensing is the mechanism by which adequate digestive or hormonal responses are ensured after ingestion of fuel substrates. Nutrient taste receptors (TASs) have been well characterized in the oral cavity, but recently they have also been found in the gut mucosa [18]. Within the gastrointestinal (GI) tract, two families of TASs, the taste 1 receptor family (TAS1R), which detects umami and sweet stimuli, and the taste 2 receptor family (TAS2R), which detects bitter stimuli, occur as shown in Figure 1 [19,20]. The activation of these GI receptors results in the release of intestinal hormones such as cholecystokinin (CKK) or glucose-dependent insulinotropic polypeptide and glucagon-like peptide-1 (GLP-1), which modulate the physiological response to nutrients, particularly satiety and the maintenance of energy homeostasis [21]. Recent studies carried out in humans and animal models with obesity and/or T2DM have demonstrated a downregulation of TASs, although the mechanisms are still unknown [22,23].

It seems that TASs could be deregulated in metabolic diseases such as obesity and T2DM, as well as in MetS or NAFLD; therefore, deepening our knowledge about the precise role of these receptors in metabolic dysfunction could be important to find new therapeutic strategies for these prevalent diseases.

In this sense, we wanted to analyze jejunal sweet and bitter TASs. While sweet sensing is achieved by the TAS1R2–TAS1R3 heterodimer [26], there are 25 TAS2Rs for bitter sensing [27]. Xie et al. summarized that TAS2R5, TAS2R14 and TAS2R38 were expressed in the small intestine [28], and TAS2R14 and TAS2R38 seemed to be involved in the release of CKK and GLP-1, respectively [29,30]. In this regard, the main objective of the present work was to focus on the study of the relative expression of the TAS1R2 and TAS1R3 sweet sensing receptors, and of the TAS2R14 and TAS2R38 bitter sensing receptors in jejunal samples of women with morbid obesity (MO) classified into the presence or absence of MetS and/or NAFLD.

## 2. Results

### 2.1. Baseline Characteristics of Subjects

The clinical characteristics and biochemical measurements of the population studied are shown in Table 1. All of our patients were women with MO at first classified into nonmetabolic syndrome (Non-MetS; *n* = 45) and MetS (*n* = 24) according to Alberti et al. criteria [31]. Both groups showed no differences in weight, body mass index (BMI), diastolic blood pressure (DBP), homeostatic model assessment method 2 of insulin resistance (HOMA2-IR), insulin, cholesterol, low density lipoprotein cholesterol (LDL-C), high density lipoprotein cholesterol (HDL-C), alanine aminotransferase (ALT), gamma-glutamyltransferase (GGT) and alkaline phosphatase (ALP). In this analysis, as expected, we observed that patients with MetS had significantly higher levels of systolic blood pressure (SBP; *p* = 0.014), glucose (*p* = 0.001), triglycerides (TG; *p* =< 0.001) and aspartate aminotransferase (AST; *p* = 0.005) than Non-MetS subjects.

To add more knowledge about the role of TAS, we wanted to analyze the clinical and biochemical variables of the same patients classified according to the absence or the presence of NAFLD. The cohort was composed of 28 women with normal liver (NL) and 41 women with NAFLD; 17 of them also presented MetS, (Table 2). The cohort of patients showed no differences in weight, BMI, SBP, DBP, HOMA2-IR, insulin, TG, cholesterol, LDL-C, HDL-C, AST, ALT or GGT. Biochemical analyses indicated that patients with NAFLD had significantly higher levels of fasting glucose (*p* = 0.026) and ALP (*p* = 0.009) than patients with NL histology.

Furthermore, we subclassified our patients with MO and NAFLD into 24 women with simple steatosis (SS) and 17 women with NASH (Table 2). Fasting glucose (*p* = 0.021), ALT (*p* = 0.042) and ALP (*p* = 0.001) levels were significantly higher in the SS group than in NL subjects. ALP (*p* = 0.008) levels were significantly higher in SS than in NASH.

### 2.2. Nutrient Taste Receptors Expression in Jejunum According to Metabolic Diseases

Given that TAS have been related to metabolic diseases such as obesity and T2DM, and our cohort of patients was composed by women with MO, at first we wanted to analyze TAS differential jejunal relative expression (JRE) according to T2DM presence, but we only had three patients with T2DM in our cohort that also presented MetS, and there were no significant differences in TAS (TAS1R3, TAS2R38 and TAS2R14) expressions (*p* = 0.074, *p* = 0.857, *p* = 0.749, respectively). It is important to note that we did not observe expression of TAS1R2 in jejunum samples of any patients of the whole cohort.

We analyzed TAS expressions according to MetS presence. In this case, we had 45 patients without MetS and 24 patients with MetS. Although we did not find differences in JRE of TAS1R3 (Figure 2A) and TAS2R38 (Figure 2C) according to this classification, we found that JRE of TAS2R14 was significantly lower in MetS group in comparison to non-MetS subjects (Figure 2B).

### 2.3. Nutrient Taste Receptors Expression in Jejunum According to NAFLD Classification

To examine the possible role of TAS in NAFLD, we wanted to evaluate the JRE of TAS1R2, TAS1R3, TAS2R14 and TAS2R38 in a cohort with MO classified into NL and NAFLD groups. As stated above, we did not observe expression of TAS1R2 in jejunum samples. Besides, we did not report differences in JRE of TAS1R3 (Figure 3A), TAS2R14 (Figure 3C) or TAS2R38 (Figure 3E) between NL group and NAFLD subjects.

When we subclassified our cohort of patients according to the hepatic histopathological classification into NL, SS and NASH, we did not observe differences in JRE of TAS1R2 (Figure 3B), TAS2R14 (Figure 3D) or TAS2R38 (Figure 3F).

### 2.4. Correlations between TAS Jejunal Expressions and Other Biochemical Parameters and Genes Related to Lipid Metabolism

To study the possible relationship between TAS and MetS and/or NAFLD, we analyzed associations with anthropometric and biochemical parameters, levels of some adipocytokines and TLRs jejunal expressions, as shown Table 3. Furthermore, we found a negative correlation between the presence of MetS with JRE of TAS1R3 (rho = −0.330) and with JRE of TAS2R14 (rho = −0.389).

## 3. Discussion

Nutrient taste receptors have been fully described in the oral cavity [32], but recent studies have demonstrated that they can also be found in GI tract [33] and are involved in satiety and homeostatic control functions [24,34]. Some authors have reported TAS downregulation in metabolic diseases such obesity and T2DM [24]. The novelty of the present study lies in the fact that we analyzed the JRE of TAS1R2, TAS1R3, TAS2R14 and TAS2R38 in a well-characterized cohort of women with MO according to the presence or absence of MetS, and according to the presence of NAFLD.

First, we want to mention that our cohort of patients did not express TAS1R2 in the jejunum. TAS1R2 forms a heterodimer with TAS1R3 to sense sweet nutrients. Curiously, although we did not observe the presence of TAS1R2, our patients expressed TAS1R3 in the jejunum. Similarly, another report did not detect TAS1R2 in the GI tract of humans or animal models, but they detected other TAS1 families [35].

To achieve the main objective of this study, we first analyzed the relationship between the JRE of TAS1R3, TAS2R14 and TAS2R38 in the presence or absence of MetS. We reported a significant decrease only in TAS2R14 expression in patients suffering from MetS compared with those without the syndrome. Our results seemed to agree with Janssen et al., who demonstrated in animal models that the activation of bitter taste regulates satiety [36], favoring energy and metabolic homeostasis. Moreover, Kok et al. demonstrated that TAS2R108 activation in the gut of animal models seems to cause changes in the bile acid metabolism and enteroendrocrine hormone release, ameliorating multiple features of metabolic syndrome [37]. However, there are no reports of bitter TASs in human subjects in the context of obesity or metabolic dysfunction to more deeply explain this result. In this sense, we postulate that the metabolic dysfunction of these MetS patients could be associated with the downregulation of TAS2R14. Further studies in human cohorts are needed to confirm this hypothesis.

To add new knowledge about the implications of TASs in NAFLD, we classified our cohort according to hepatic histology into NL and NAFLD. Then, we subclassified our NAFLD patients according to hepatic histopathology as SS and NASH subjects. Unfortunately, regarding this classification, we did not find significant differences in the JRE of TAS1R3, TAS2R14 or TAS2R38 among the groups. This is one of the main novelties of the present study, since there are no previous works on this topic.

Additionally, we evaluated some associations between the JRE of TAS1R3, TAS2R14 and TAS2R38, and some parameters involved in MetS and NAFLD pathogenesis. In this sense, we found a negative correlation between glucose and TG with TAS1R3, which is explained because this receptor senses glucose and controls its absorption [24,32,33]. Moreover, Widmayer et al. reported that patients with obesity presented with reduced levels of JRE of TAS1R3 [38]. In addition, MetS correlates negatively with the JRE of TAS1R3 and TAS2R14, from which it could be inferred that patients with MetS have severe metabolic dysfunction that could deregulate TASs.

As systemic inflammation underlies MetS and NAFLD, we also studied the correlation between jejunal TAS expression with some adipocytokine levels, and we observed interesting significant associations. In this sense, we found a positive correlation between IL-10, an anti-inflammatory adipocytokines, and the JRE of TAS2R14, and a negative correlation between IL-6, a proinflammatory adipocytokine, and the JRE of TAS2R38. It has been reported that obese patients present with chronic low-grade inflammation and downregulation of adipocytokine levels in their blood [39,40]. These facts agree with our results, since our cohort was made up of patients with MO, and it seems that this chronic low-grade inflammation reduces the expression of bitter receptors, the same as Widmayer et al. reported for the sweet receptor TAS1R3 [38].

Moreover, we reported that ghrelin levels were negatively associated with the JRE of TAS2R14. This result could be explained by diet, since some studies reported that ghrelin release is dependent on nutritional state and diet composition [41,42]. In this regard, our patients with MO underwent a very low-calorie diet three months before bariatric surgery, which may have modulated ghrelin levels before blood extraction.

Furthermore, JRE of peroxisome proliferator-activated receptor γ (PPAR-γ) and TLR4 correlated positively with TAS2R14. Additionally, TAS2R38 correlated negatively with JRE of TLR9 and positively with circulating TLR4 levels. To explain these results, on the one hand, the PPAR-γ association agrees with Yajima et al., who demonstrated that chemical compounds contribute to bitter taste, improving insulin sensitivity by activating PPAR in high fat diet-induced obese mice and in patients with T2DM [43]. On the other hand, curiously, the correlations between TLR and TASs are very controversial. Previous reports have shown that TLRs induce inflammation under certain conditions, such as obesity or NAFLD [44,45,46]. Accordingly, our patients with MO showed enhanced levels of proinflammatory cytokines. Additionally, the negative association between the JRE of TLR9 and TAS2R38 was in agreement with the literature [35,37]. However, the correlation between TASs and TLR4 is controversial, probably due to the small number of studied patients. In this sense, further studies are needed to clarify the relationship between TAS and TLR.

In the current study, we reported for the first time a relationship between jejunal TAS expression and MetS without the interference of confounding factors such as sex or age. Although these are preliminary results, they seem to suggest that taste receptor modulation could be a possible therapeutic target for metabolic disorders. However, further human studies are needed to corroborate this relationship.

The main limitation of our work was that we assessed the whole study in a specific cohort of women with MO. Hence, these results cannot be extrapolated to men, women of other ages or individuals with overweight or normal weight. Additional studies would be useful to validate these findings.

## 4. Materials and Methods

### 4.1. Study Subjects

The study (23c/2015) was approved by the institutional review board “Comitè d’Ètica d’Investigació Clínica, Hospital Universitari Joan XXIII de Tarragona”. Sixty-nine Caucasian women with MO (BMI > 40 kg/m^2^) who underwent a metabolic and bariatric Roux-en-Y gastric bypass (RYGB) were included in the studied cohort. All of them gave a signed informed consent to participate in the research. Hepatic and jejunal biopsies were obtained during the surgery. To diagnose NAFLD in the participants, the following criteria were considered: (1) liver pathology, (2) an intake of less than 10 g of ethanol/day and (3) the exclusion of other liver diseases. The study exclusion criteria were: (1) an intake of ethanol higher than 10 g/day or other toxins; (2) subjects who had an acute or chronic hepatic or inflammatory disease, infectious disease or neoplastic disease; (3) menopausal women or women using contraceptives; (4) women with diabetes receiving insulin or another medication that can modulate endogenous insulin levels and (5) patients treated with fibrates.

According to the hepatic histopathological classification described elsewhere [47], women with MO who followed the study criteria were included in the research and were subclassified by an experienced pathologist into obese patients with NL (*n* = 28) and NAFLD (*n* = 41) [SS (micro/macrovesicular steatosis without inflammation or fibrosis, *n* = 24) and NASH (Brunt Grades 1–2, *n* = 17)]. None of the NASH patients included had fibrosis.

### 4.2. Sample Size

In accordance with GRANMO Sample Size Calculator, accepting an α risk of 0.05 and a β risk of less than 0.2 in a bilateral contrast, 24 subjects per group are needed to detect a difference ≥ 0.2 units. It was assumed that the common standard deviation was 0.3.

### 4.3. Biochemical Analyses

The entire studied cohort underwent physical, anthropometric and biochemical assessments. Specialized nurses performed blood extraction using a BD Vacutainer^®^ system after overnight fasting before gastric bypass. Serum and plasma samples were obtained using empty and ethylenediaminetetraacetic acid coated tubes, and a subsequent centrifugation (3500 rpm, 15 min, 4 °C) was performed. A conventional automated analyzer was used to analyze the biochemical variables. Insulin resistance was estimated using HOMA2-IR.

Plasma and serum samples were stored at −80 °C. Themultiplex sandwich immunoassays, the MILLIPLEX MAP Human Adipokine Magnetic Bead Panel 1 (HADK1MAG-61K, Millipore, Billerica, MA, USA), the MILLIPLEX MAP Human High-Sensitivity T Cell Panel (HSTCMAG28SK, Millipore, Billerica, MA, USA), and the Bio-Plex 200 instrument at the Center for Omic Sciences (Universitat Rovira i Virgili) were used to determine the circulating levels of IL-1β, IL-6, IL-7, IL-8 and TNF-α, according to the manufacturer’s instructions. Circulating levels of IL-17, IL-10 and IL-22 (Quantikine, R&D Systems, Minneapolis, MN, USA) were measured in duplicate using enzyme-linked immunosorbent assay (ELISA) following the protocol. TLR2, TLR4 and TLR9 levels were analyzed by enzyme-linked immunosorbent assay (ELISA) according to the manufacturer’s protocol (Ref. SEA753Hu; USCN). Ghrelin was also detected by ELISA (Linco Research, Saint Charles, MO, USA).

### 4.4. Jejunal Gene Expression

Jejunum samples were collected at surgery and conserved in RNAlater (Qiagen, Hilden, Germany) tubes, conserved at 4 °C and then processed and stored at −80 °C. An RNeasy mini kit (Qiagen, Barcelona, Spain) allowed extraction of total RNA from the tissue samples. Reverse transcription to cDNA was performed with the High-Capacity RNA-to-cDNA Kit (Applied Biosystems, Madrid, Spain). Real-time quantitative PCR was carried out with the TaqMan Assay predesigned by Applied Biosystems for the detection of TAS1R2 (Hs01027711_m1), TAS1R3 (Hs00877446_g1), TAS2R14 (Hs00256800_s1), TAS2R38 (Hs00604294_s1), PPAR-α (Hs00947538_m1), PPAR-γ (Hs01115513_m1), TLR2 (Hs02621280_s1), TLR4 (Hs00152939_m1) and TLR9 (Hs00370913_s1). The expression of each gene was calculated relative to the expression of glyceraldehyde-3-phosphate dehydrogenase (Hs02786624_g1), a housekeeping gene, and was normalized using NL subjects as a reference control group. All reactions were performed in duplicate in 96-well plates using the 7900HT Fast Real-Time PCR system (Applied Biosystem, Foster City, CA, USA). The final PCR volume was 20 μL. Cycling conditions were as follows: 95 °C for 2 s, followed by 40 cycles of 95 °C for 1” and 60 °C for 20”.

### 4.5. Statistical Analysis

The SPSS/PC+ for Windows statistical package was used to analyze the data (version 23.0; SPSS, Chicago, IL, USA). The Kolmogorov-Smirnov test was used to assess the distribution of variables. Continuous demographic, clinical and laboratory measures are reported as means ± SD if they were parametric variables, or as medians and 25–75th percentiles if they were nonparametric variables. To compare the difference between two or more groups, a nonparametric Mann-Whitney’s U test or Kruskal–Wallis test was performed. The strength of the association between variables was calculated using Pearson’s method (parametric variables) and Spearman’s ρ correlation test (nonparametric variables). *p*-values < 0.05 were statistically significant.

## 5. Conclusions

In conclusion, our findings suggest that metabolic dysfunctions such as MetS trigger downregulation of the intestinal TASs, causing altered nutrient sensing that in the long term could aggravate metabolic disorders. Therefore, modulation of these receptors could be a possible therapeutic target to improve metabolic homeostasis.

## Figures and Tables

**Figure 1 nutrients-13-02437-f001:**
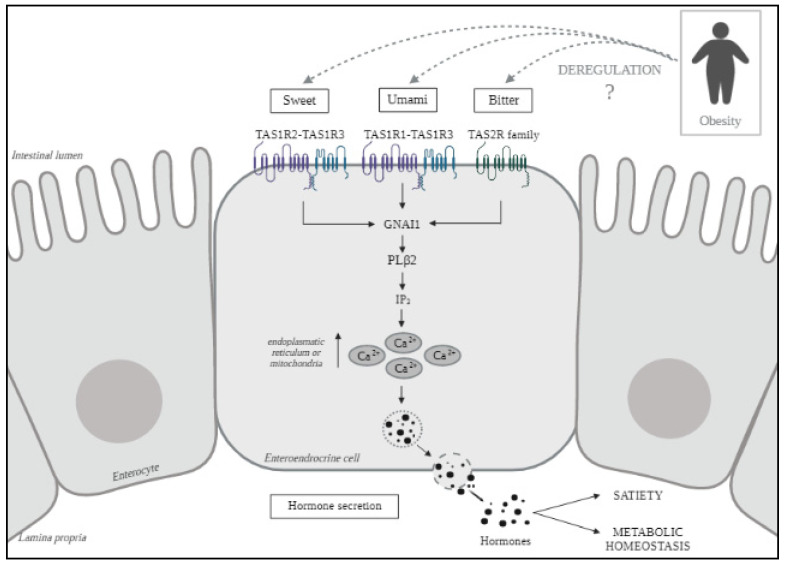
Nutrient taste receptors signaling in enteroendrocrine cells. The detection of sugars molecules is mediated by TAS1R2–TAS1R3, umami sensing is achieved by TAS1R1–TAS1R3 and bitter detection is mediated by the TAS2R family. When nutrients bind their receptors, the signaling pathway is activated, causing an increase of intracellular calcium that ultimately induces the release of hormones involved in the process of satiety and metabolic homeostasis [24]. Obesity seems to deregulate this pathway but the precise mechanism is still unknown [25]. TAS1, taste 1 receptor family; TAS2, taste 2 receptor family; GNAI1, guanine nucleotide-binding protein G; PLβ2, phosphatidylinositol phospholipase C; IP_3_, inositol trisphosphate; Ca^2+^, calcium ^2+^.

**Figure 2 nutrients-13-02437-f002:**
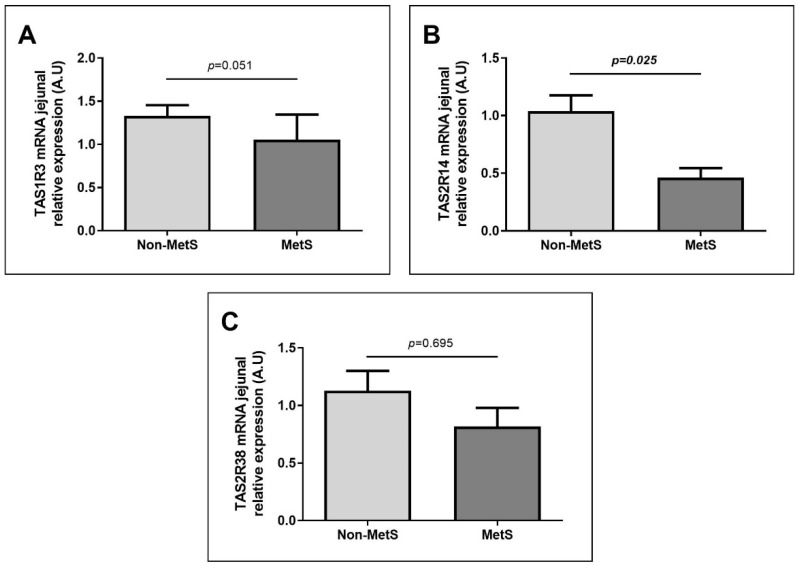
Differential JRE of TAS in women with morbid obesity classified according to the presence or absence of MetS (**A**) TAS1R2 JRE of women with or without MetS. (**B**) TAS2R14 JRE of women with or without MetS. (**C**) TAS2R38 JRE of women with or without MetS A.U, arbitrary units; Non-MetS, non-metabolic syndrome; MetS, metabolic syndrome; TAS1R3, taste 1 receptor family 3; TAS2R14, taste 2 receptor family 14; TAS2R38, taste 2 receptor family 38. *p* < 0.05 was considered statistically significant.

**Figure 3 nutrients-13-02437-f003:**
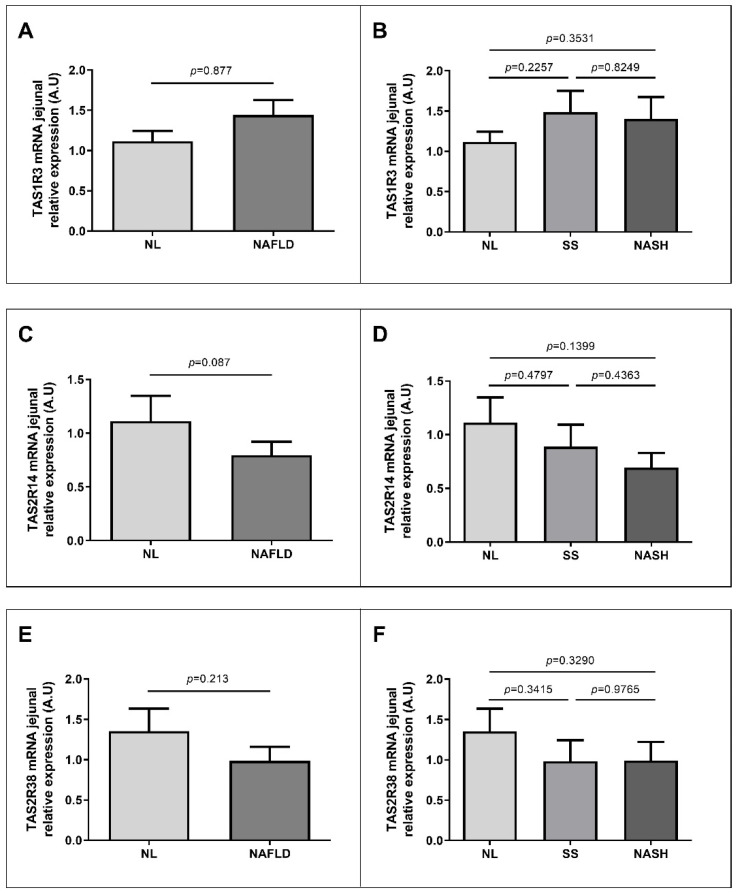
Differential JRE of TAS in women with morbid obesity classified according to hepatic histopathological classification. (**A**) TAS1R3 JRE of NL and NAFLD subjects; (**B**) TAS1R3 JRE of NL, SS and NASH groups; (**C**) TAS2R14 JRE of NL and NAFLD subjects; (**D**) TAS2R14 JRE of NL, SS and NASH groups; (**E**) TAS2R38 JRE of NL and NAFLD subjects; (**F**) TAS2R38 JRE of NL, SS and NASH groups. TAS, taste receptors; NL, normal liver; NAFLD, nonalcoholic fatty liver disease; SS, simple steatosis; NASH, nonalcoholic steatohepatitis; TAS1R3, taste 1 receptor family 3; TAS2R14, taste 2 receptor family 14; TAS2R38, taste 2 receptor family 38; A.U, arbitrary units. *p* < 0.05 was considered statistically significant.

**Table 1 nutrients-13-02437-t001:** Clinical and biochemical variables of women classified according to the presence or absence of metabolic syndrome.

Variables	Non-MetS	MetS
	(*n* = 45)	(*n* = 24)
Weight (kg)	117.00 (108.00–128.50)	112.00 (105.48–125.50)
BMI (kg/m^2^)	44.14 (41.34–46.63)	42.64 (40.67–46.24)
SBP (mmHg)	114.39 ± 13.86	124.00 ± 16.55 *
DBP (mmHg)	62.50 (59.00–71.25)	63.50 (56.50–74.75)
HOMA2-IR	1.10 (0.79–1.66)	1.35(0.77–3.57)
Glucose (mg/dL)	85.00 (76.00–93.00)	100.50 (89.75–109.00) *
Insulin (mUI/L)	8.49 (6.00–13.25)	10.22 (5.68–32.41)
TG (mg/dL)	106.00 (77.50–132.00)	168.50 (124.00–232.25) *
Cholesterol (mg/dL)	168.72 ± 33.02	186.58 ± 41.65
LDL-C (mg/dL)	102.37 ± 25.79	110.68 ± 33.98
HDL-C (mg/dL)	43.44 ± 11.93	37.77 ± 7.26
AST (UI/L)	19.50 (15.25–30.75)	33.00 (20.75–45.75) *
ALT (UI/L)	23.00 (17.00–34.50)	34.00 (23.00–43.00)
GGT (UI/L)	20.00 (14.50–27.00)	22.00 (16.00–29.50)
ALP (Ul/L)	66.48 ± 15.10	67.57 ± 12.07

Non-MetS, nonmetabolic syndrome; MetS, metabolic syndrome; BMI, body mass index; SBP, systolic blood pressure; DBP, diastolic blood pressure; HOMA2-IR, homeostatic model assessment method 2 of insulin resistance; TG, triglycerides; LDL-C, low density lipoprotein cholesterol; HDL-C, high density lipoprotein cholesterol; AST, aspartate aminotransferase; ALT, alanine aminotransferase; GGT, gamma-glutamyltransferase; ALP, alkaline phosphatase. Data are expressed as the mean ± standard deviation or median (interquartile range), depending on the distribution of the variables. * Significant differences between Non-MetS group and MetS group (*p* < 0.05).

**Table 2 nutrients-13-02437-t002:** Clinical and biochemical variables of women classified according to the hepatic histopathological classification.

Variables	NL	NAFLD	SS	NASH
	(*N* = 28)	(*N* = 41)	(*N* = 24)	(*N* = 17)
Weight (kg)	116.50 (107.25–130.50)	112.40 (106.00–128.00)	113.20 (108.33–128.00)	112.00 (104.65–125.00)
BMI (kg/m^2^)	43.30 (40.94–46.47)	44.46 (40.84–46.60)	44.35 (40.82–46.83)	44.46 (40.76–46.03)
SBP (mmHg)	119.00 ± 18.26	117.29 ± 13.86	120.09 ± 13.41	113.44 ± 13.96
DBP (mmHg)	63.00 (57.75–75.75)	62.00 (59.00–71.25)	62.00 (59.00–72.50)	65.50 (56.75–70.75)
HOMA2-IR	1.23 (0.75–2.05)	1.25 (0.79–2.18)	1.49 (0.95–2.18)	0.86 (0.61–3.00)
Glucose (mg/dL)	85.50 (76.25–93.00)	93.00 (84.00–105.00) *	93.50 (85.75–107.00) ^#^	91.00 (82.50–101.20)
Insulin (mUI/L)	9.43 (5.59–16.21)	9.63 (5.88–14.52)	11.27 (7.81–14.51)	6.57 (5.09–23.04)
TG (mg/dL)	106.00 (89.00–136.00)	132.00 (91.00–189.00)	129.50 (85.75- 175.50)	140.00 (106.00–247.00)
Cholesterol (mg/dL)	171.88 ± 36.20	179.07 ± 38.80	174.42 ± 35.41	185.28 ± 43.39
LDL-C (mg/dL)	108.16 ± 27.94	104.48 ± 30.86	104.39 ± 31.21	104.62 ± 31.58
HDL-C (mg/dL)	40.84 ± 9.89	41.04 ± 10.95	42.56 ± 12.38	38.89 ± 8.47
AST (UI/L)	20.50 (15.75–36.25)	23.50 (17.00–41.75)	23.00 (17.00–35.00)	24.00 (17.00–43.00)
ALT (UI/L)	22.00 (16.00–27.00)	31.00 (21.00–37.00)	31.00 (23.00–35.75) ^#^	30.00 (15.50–40.00)
GGT (UI/L)	18.00 (16.00–27.00)	22.00 (16.00–27.00)	21.00 (16.25–30.50)	25.00 (15.00–27.00)
ALP (Ul/L)	60.42 ± 13.09	70.67 ± 13.01 *	75.80 ± 11.66 ^#^	62.77 ± 11.16 ^&^

NL, normal liver; NAFLD, nonalcoholic fatty liver disease; SS, simple steatosis; NASH, nonalcoholic steatohepatitis; BMI, body mass index; SBP, systolic blood pressure; DBP, diastolic blood pressure; HOMA2-IR, homeostatic model assessment method 2 of insulin resistance; TG, triglycerides; LDL-C, low density lipoprotein cholesterol; HDL-C, high density lipoprotein cholesterol; AST, aspartate aminotransferase; ALT, alanine aminotransferase; GGT, gamma-glutamyltransferase; ALP, alkaline phosphatase. Data are expressed as the mean ± standard deviation or median (interquartile range), depending on the distribution of the variables. * Significant differences between NL group and NAFLD group (*p* < 0.05). ^#^ Significant differences between NL group and SS group (*p* < 0.05). ^&^ Significant differences between SS group and NASH group (*p* < 0.05).

**Table 3 nutrients-13-02437-t003:** Significant correlations between different parameters and TAS1R3, TAS2R14 or TAS2R38 jejunal relative expression in the whole cohort of studied subjects.

Variables	TAS1R3 JRE	TAS2R14 JRE	TAS2R38 JRE
Glucose (mg/dL)	−0.414 *	ns	ns
TG (mg/dL)	−0.608 **	ns	ns
JRE TLR4	ns	0.472 **	ns
JRE TLR9	ns	ns	−0.481 *
JRE PPAR-γ	ns	0.364 *	ns
IL-6 (pg/mL)	ns	ns	−0.434 *
TLR4 (ng/mL)	ns	ns	0.410 *
Ghrelin (ng/mL)	ns	−0.900 *	ns
IL-10 (pg/mL)	ns	0.481 **	ns

TAS1R3, taste 1 receptor family 3; TAS2R14, taste 2 receptor family 14; TAS2R38, taste 2 receptor family 38; JRE, jejunal relative expression; ns, nonsignificant correlations; TG, triglycerides; MetS, metabolic syndrome; TLR, toll-like receptor; PPAR-γ, peroxisome proliferator-activated receptor γ; IL, interleukin. Data are expressed as the correlation coefficient rho of Spearman and *p*-value (*p* < 0.05 was considered statistically significant; * *p* < 0.05, ** *p* < 0.01).

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
