# Peer review of "Expression of Jejunal Taste Receptors in Women with Morbid Obesity"

_nutrients, 2021, doi:10.3390/nu13072437_

Round 1

Reviewer 1 Report

  • Why the study focuses only on women?
  • Please add a reference to this sentence: "Nutrient taste receptors (TASs) have been well characterized in the oral cavity, but recently, they have also been found in the gut mucosa."
  • Which program was used to calculate the minimum sample size?
  • Please provide a PCR condition in the Methods section.
  • Which criteria were used to diagnose metabolic syndrome?
  • Please consider conducting separate correlations in the group of women with and without metabolic syndrome and with and without NAFLD / NASH.
  • What are the strengths and limitations of this study?

Reviewer 2 Report

The study described in the manuscript titled “Deregulation of jejunal taste receptors in metabolic dysfunction associated diseases in women with morbid obesity” is very interesting, well designed and well conducted.

However, there is one point that needs being carefully elucidated before publication.

  • The main objective of the study was to analyze the relative expression of TAS1R2, TAS1R3, TAS2R14 and TAS2R38 in jejunal samples of women with morbid obesity. Since it is known that humans possess ~25 TAS2R bitter receptors, Authors should explain the reason way they selected TAS2R14 and TAS2R38. Are these two TAS2Rs the only ones known to be expressed in the gastrointestinal tract? Please add this in the introduction.
  • In table 1 Authors must specify what the numbers indicate. Are the numbers the Means ± standard errors? What do the numbers in brackets represent?
  • in Table 3 the correlations are shown between TAS jejunal expressions and other biochemical parameters and genes related to lipid metabolism. Did the Authors report only the significant associations found with variables and JRE or TASs? If so, please specify in the text and briefly describe the significant correlations that you found.
  • Since Authors reported that patients with MetS have a higher risk of developing type 2 diabetes mellitus (T2DM) or nonalcoholic fatty liver disease NAFLD, it would be important to include the number of patients with MetS who also have NAFLD or T2DM. I would suggest including this information in a table showing the characteristics of the patients.
  • In the Sample size section please indicate how the sample size was calculated.
  • In the Jejunal gene expression section Authors should indicate how they performed the expression of TAS genes.
  • In the abstract, at line 25, use for the first time the abbreviation NAFLD; please add: “(nonalcoholic fatty liver disease)”; in the results section, at line 123, please define the abbreviation NASH (nonalcoholic steatohepatitis).

Round 2

Reviewer 1 Report

The authors have successfully addressed most of my concerns.

Author Response

Thank you for your evaluation.